# Improving the Rheological Properties of Dough Obtained by Partial Substitution of Wheat Flour with Freeze-Dried Olive Pomace

**DOI:** 10.3390/foods13030478

**Published:** 2024-02-02

**Authors:** Patricia Dahdah, Roberto Cabizza, Maria Grazia Farbo, Costantino Fadda, Andrea Mara, Georges Hassoun, Antonio Piga

**Affiliations:** 1Dipartimento di Agraria, Università degli Studi di Sassari, Viale Italia 39, 07100 Sassari, Italy; pdahdah@uniss.it (P.D.); rcabizza@uniss.it (R.C.); mgfarbo@uniss.it (M.G.F.); cfadda@uniss.it (C.F.); 2Dipartimento di Scienze Chimiche, Fisiche, Matematiche e Naturali, Università degli Studi di Sassari, Via Vienna 2, 07100 Sassari, Italy; a.mara@studenti.uniss.it; 3Department of Environment and Natural Resources, Lebanese University, Beirut P.O. Box 6573/14, Lebanon; hassoungeorges@gmail.com

**Keywords:** dough rheology, fibre content, gluten network, olive pomace, polyphenol content, wheat dough

## Abstract

Mediterranean countries are known for their high-quality olives and the production and consumption of olive oil. Olive pomace (OP), the major by-product of olive oil extraction, is receiving attention for its potential as a functional compound in food products, reflecting its physiology- and health-promoting attributes. This study assessed the physico-chemical characteristics of OP obtained from two Sardinian olive cultivars, Bosana and Semidana, and the effect of OP incorporation on the baking performance of wheat dough. We assessed the rheological parameters, pasting profile, and fermentation of doughs obtained through the partial substitution of wheat flour with OP at 0 (control), 1, 2, 3, and 5%. OP inclusion resulted in significant differences in the studied parameters compared with control samples. Positive effects included a decrease in development time, improved dough stability and storage, and superior loss modulus and gas retention capacity. Negative effects comprised an increase in dough resistance and a decrease in dough development height, gas production, gas retention, pasting profile, stickiness, and elasticity. These differences in the OP dough were due to the interactions between polyphenols and fibre with water and the starch–gluten matrix. This study found improvements in dough characteristics following the substitution of wheat flour with low percentages of OP, especially Semidana at 1%. Although higher percentages of OP would be associated with greater nutritional and health benefits, they resulted in a degradation of the dough’s attributes, producing a gluten-free-like matrix in the final product.

## 1. Introduction

Agro-food industry wastes are a matter of great concern due to the environmental and economic impacts their disposal has on society. Finding alternative uses for these by-products can ameliorate these effects. For instance, many products could be directed back into the food chain because of their capacity to confer health benefits when consumed, for example, by contributing towards the prevention of cancer, inflammatory disorders, and cardiovascular diseases [1]. Considering the fact that waste products arising from the agro-food industry are generally cheap and offer sources of an important number of nutraceuticals and functional compounds, such as polyphenols and fibre, the incorporation of these by-products into foods can create new products that are both more sustainable and nutritious [2,3]. However, the incorporation of functional compounds into foods can be challenging for manufacturers because it may negatively affect the product’s quality, and, thus, consumer acceptance, for example, by decreasing bread loaf volume or resulting in undesirable textural and sensory characteristics. This raises the need to find a balance between including functional compounds in recipes whilst causing minimal side effects on the product quality [4].

Considering the high consumption of cereal products worldwide, many trials have already been carried out [5] focusing on the incorporation of agro-food by-products into dough preparations [5]. Potential by-products include apple pomace [6], orange peel powder [2], tomato pomace powder [7], rosehip seed flour [8], carrot pomace [9], grape seed flour [4,10], maize bran [11], and grape pomace [12]. For example, the high fibre content of orange peel powder was associated with an improved water-binding capacity, water retention capacity, and the viscous properties of bread dough [2].

In Mediterranean countries, an important source of by-products derives from the olive oil extraction industry. Being a cheap raw material and a source of antioxidant and fibre compounds, olive oil by-products have the potential to be used in various applications such as food, nutraceuticals, animal feed, and cosmetics. Indeed, the rich source of compounds found in olives offers an excellent opportunity to transform an agricultural waste product into useful, functional ingredients. However, incorporating olive oil by-products into food products may affect their sensory and technological characteristics. As a result, determining the appropriate amounts to add requires careful evaluation. In fact, the challenge of keeping sensory and technological characteristics within acceptable ranges may significantly limit the use of olive oil by-products for food fortification [5,13].

Olive pomace (OP) is a major by-product of the olive oil industry. It constitutes around 65% of the initial weight of pressed olives, considering a three-phased pressing system, or 80% of the initial weight, considering a two-phased decanter [14,15]. It is made up of approximately 20% (*w*/*w*) olive husks and pulp and 15% (*w*/*w*) crushed olive stones, plus the residual oil and water from the olive oil extraction process [16].

Phenolic compounds are the main antioxidant elements found in OP. The principal phenolic compounds are hydroxytyrosol, oleuropein, tyrosol, caffeic acid, p-coumaric acid, vanillic acid, verbascoside, elenolic acid, catechol, and rutin [17]. Upon consumption, most are hydrolysed into hydroxytyrosol, the absorption of which takes place in the small intestine. Hydroxytyrosol and its derivatives are responsible for the “healing effect” of the polyphenols present in olive oil and OP on the human body [18,19,20]. Several studies provide evidence showing antioxidants to slow biological processes such as aging, and to protect against several diseases such as cancer, cardiovascular diseases, and diabetes [1,5,18,21,22]. Antioxidants defend cellular processes from free radicals, the molecules responsible for oxidative stress. Recent studies have also demonstrated dietary polyphenols from olive products to protect against other detrimental mechanisms and processes, for example, by participating in the activation of signalling pathways involved in the prevention of inflammation, oxidative stress, and insulin resistance [18,23]. When used as natural food additives, and, due to their antioxidant activity, polyphenols extend the shelf-life of food and help to reduce nutritional loss and the formation of harmful substances [18,24]. This effect is particularly important since their use is targeted to replace synthetic, harmful antioxidants such butylated hydroxyanisole (BHA) and butylated hydroxytoluene (BHT). OP was also identified as an important source of fatty acids—mainly oleic, palmitic, and linoleic acids [18].

Considering the health benefits of OP, the possibility of incorporating it into food is both logical and sustainable [18]. In this context, we studied the OPs derived from two Sardinian olive cultivars, Bosana and Semidana, following extra virgin olive oil production, and assessed their effects on the rheological characteristics and nutritional value of dough destined for breadmaking.

## 2. Materials and Methods

### 2.1. Sampling

Raw olive pomace (ROP) from the Sardinian olive cultivars Bosana and Semidana were collected using a LEOPARD multiphase decanter (DMF) (Pieralisi, Jesi, Italy), at Accademia Olearia (Alghero, Italy), during the 2021–2022 harvesting season. The DMF is an evolution of the two-phase decanter. It is an automated system able to operate in both continuous and batch modes. Similar to the two-phased system, the DMF does not require additional water in the malaxation step. This decanter has three outlets for collecting the oil phase, the dry solid kernel-enriched phase, and the wet semi-solid phase, thus permitting the separation of the pulp from the seed, and the collection a seedless wet pulp known as patè [25,26,27,28,29]. The dry solid phase is similar to the one obtained from three-phase decanters, with an average moisture content of 45–55%, while the pitted wet OP obtained has an average moisture content of 75–90% [30]. ROP samples were immediately frozen and stored at −20 °C until analysis.

An aliquot of each sample was freeze-dried for 48 h using a Labconco 8L −50 °C series freeze dryer (Kansas City, MO, USA) at a collecting temperature of −54.0 °C and a pressure of 0.1 mbar. The samples were then ground to a particle size < 500 µm using a domestic grinder (Moulinex A320R1, 700 W, Paris, France) to produce the final freeze-dried OP (FD-OP), and stored under vacuum conditions in the dark at room temperature until analysis.

### 2.2. Characterization of Olive Pomace

The moisture content of ROP samples was assessed by exposing the samples to a temperature of 105 °C in an oven until a constant weight was achieved [31]. Ash content was evaluated using a muffle furnace set to 550 °C [32]. Lipid content was assessed following the AOAC 2003.06–2006 official method [33], using a solvent extractor (SER 158, Velp Scientific, Deer Park, NY, USA) and petroleum ether (Carlo Erba Reagents, Milan, Italy) as solvent. Lipid content was quantified gravimetrically.

The protein content of ROP was determined as total nitrogen content, using the Kjeldahl method and a nitrogen conversion factor of 6.25 [34].

Water activity (a_w_) was determined on homogenised ROP samples using an electronic hygrometer (A_w_-Win, Rotronic, equipped with a Karl-Fast probe, Bassersdorf, Switzerland) calibrated with known a_w_ solutions of LiCl in the range of 0.1–0.95 [35].

Carbon (C), hydrogen (H), and nitrogen (N) quantification was conducted on FD-OP using a CHN628 carbon/nitrogen/hydrogen elemental analyser (Leco Corporation, St. Joseph, MI, USA). First, 100 mg of freeze-dried, ground OP was placed in a tin capsule and sealed tightly. The capsule was then loaded into the elemental analyser and heated to high temperatures in the presence of oxygen, where the conversion of C, H, and N into gaseous CO_2_, H_2_O, and N_2_ occurs. C and H were carried by He and detected using an IR detector, while N was carried by He through a copper column and determined using a conductivity detector. The amounts of C, H, and N in the sample were calculated based on the areas of the gases produced during combustion. Instrument calibration was performed using LCRM (Leco Certified Reference Materials) [36].

Total dietary fibre (TDF) was determined gravimetrically following enzymatic digestion using the K-TDFR analytical kit (Megazyme Ltd., Bray, Ireland) [37,38]. Enzymatic digestion began by adding 50 mL of a 0.08 M phosphate-buffered solution, pH 6, to 1 g of freeze-dried OP (FD-OP). The mixture was adjusted to pH 6, then 50 µL of heat-stable α-amylase solution were added and the mixture incubated at 100 °C for 30 min. Once cool, the mixture’s pH was adjusted to 7.5 with the addition of 10 mL 0.275 N NaOH (Sigma-Aldrich, Milan, Italy). Then, 100 µL of protease solution was added, followed by a second incubation at 60 °C for 30 min. After being left to cool, the mixture was acidified to pH 4.5 with 10 mL 0.325 N HCl (Sigma-Aldrich, Milan, Italy); then, 200 µL amyloglucosidase added. The mixture was incubated, once again, at 60 °C for 30 min. Finally, 280 mL of 95% EtOH (EtOH/H_2_O 95:5 (*v*/*v*), Sigma-Aldrich, Milan, Italy), pre-heated to 60 °C, were added and the mixture left to precipitate at room temperature. The filtration took place by applying suction under vacuum conditions to crucibles containing 1 g of celite and transferring the residues intermittently. The crucibles were then subjected to washes with three parts of 20 mL of 78% EtOH (EtOH/H_2_O 78:22 (*v*/*v*)), two parts of 10 mL of 95% EtOH, and two parts of 10 mL of pure acetone (Sigma-Aldrich, Milan, Italy). The crucibles were dried in an oven at 105 °C overnight, then incinerated.

Extraction of the phenolic portion from OP was performed as reported by Simonato et al. [39] by adding 25 mL 80% MeOH (MeOH/H_2_O 80:20 (*v*/*v*), Carlo Erba Reagents, Milan, Italy) to 1 g of FD-OP. The mixture was left under magnetic stirring in the dark for 1 h at 70 °C. Afterwards, the mixture was centrifuged at 3500× *g* for 10 min at 4 °C, and the supernatant collected was used to quantify the total phenolic content (TPC) according to the methods reported by Taghouti et al. (2018) [40], with some modifications. Briefly, 7.5 mL of reverse-osmosis water (RO) were added to 1 mL of extract, plus 0.5 mL of Folin–Ciocalteu’s reagent (Carlo Erba Reagents, Milan, Italy). The mixture was vortexed, left in the dark for 5 min, and then 1 mL 5% Na_2_CO_3_ (*w*/*v*) (Carlo Erba Reagents, Milan, Italy) was added, after which the mixture was vortexed again and left in the dark at room temperature for 1 h. Absorbance was then read at 725 nm using a spectrophotometer (Cary 3500, Agilent, Cernusco, Milan, Italy). A standard curve was created using different concentrations of gallic acid (Carlo Erba Reagents, Milan, Italy) (0.01–1 mg mL^−1^). The results are expressed as gallic acid equivalents (g GAE kg^−1^ DM of FD-OP).

The antioxidant activity of OP was assessed by adding 950 µL of 0.8 mM 2,2-diphenyl-1-picrylhydrazyl solution (DPPH, Sigma-Aldrich, Milan, Italy) to 50 µL of extract in spectrophotometry cuvettes. After incubation in the dark for 30 min, absorbance was read at 517 nm [41,42,43]. Quantification was conducted using a Trolox (Sigma-Aldrich, Milan, Italy) standard curve, and the results are expressed as a µmol Trolox equivalent (TE) g^−1^ of dry matter (DM) of FD-OP.

The colour attributes of each FD-OP sample were determined using a colorimeter equipped with a CR300 measuring head (Minolta CR-300, Konica Minolta Sensing, Ramsey, NJ, USA) and applying the CIELAB (L*, a*, b*) colour system.

### 2.3. Elemental Analysis of FD-OP and Bread Samples

#### 2.3.1. Elemental Analysis of FD-OP and Bread Samples

In all the analytical phases, type I water (resistivity > 18 MΩ cm^−1^), produced using a MilliQ plus System (Millipore, Vimodrone, Italy), was used. Nitric acid (67–69% (*w*/*v*), NORMATON^®^ for ultra-trace analysis) and hydrogen peroxide (30% (*w*/*v*), NORMATON^®^ for ultra-trace metal analysis) were from VWR (Milan, Italy). The multi-element standard periodic table mix 1 (10 mg L^−1^ of Al, As, Ba, Be, Bi, B, Ca, Cd, Cs, Cr, Co, Cu, Ga, In, Fe, Pb, Li, Mg, Mn, Ni, P, K, Rb, Se, Si, Ag, Na, Sr, S, Te, Tl, V, and Zn) and mix 3 (50 mg L^−1^ of Sc, Y, La, Ce, Pr, Nd, Sm, Eu, Gd, Tb, Dy, Ho, Er, Tm, Yb, and Lu) for ICP were from Sigma-Aldrich (St. Louis, MI, USA). The single elemental standards for Na (1000 mg dm^−3^), K (1000 mg dm^−3^), Ca (1000 mg dm^−3^), Mg (1000 mg dm^−3^), P (100 mg dm^−3^), Hg (10 mg dm^−3^), Mo (10 mg dm^−3^), Sb (1000 mg dm^−3^), and Sn (100 mg dm^−3^) were from Carlo Erba (Milan, Italy). NexION KED Setup Solution (1% HCl (*v*/*v*) aqueous solution containing Co 10 μg dm^−3^ and Ce 1 μg dm^−3^) and NexION Setup Solution (1% HNO_3_ (*v*/*v*) aqueous solution containing 1 μg dm^−3^ each of Be, Ce, Fe, In, Li, Mg, Pb, and U) were from Perkin Elmer (Milano, Italy). Nylon filters (pore diameter: 0.22 μm), polypropylene (PP) metal-free tubes, and polyethene (PE) flasks were from VWR (Milan, Italy).

#### 2.3.2. Instrumentation

Trace and toxic elemental analyses were performed on a NexION 300X ICP-MS (inductively coupled plasma mass spectrometry) spectrometer equipped with an S10 autosampler, a glass concentric nebuliser, a glass cyclonic spray chamber, and a kinetic energy discrimination (KED) collision cell, all produced by Perkin Elmer (Milan, Italy). Macro elemental (Na, Mg, K and Ca) analysis was performed on an AAS (Atomic Absorption Spectrometry) analyst 200 spectrometer, produced by Perkin Elmer (Milan, Italy). The analysis of total phosphorus was carried out using the colorimetric method, by reacting an aliquot of the mineralised sample extract with phosphomolybdic reagent reduced with ascorbic acid. The colouration, developed from the reduced phosphomolybdic reagent reaction, was then measured using an Agilent Cary 60 fibre optic spectrophotometer, referring to a previously prepared calibration curve. Samples were digested using a microwave single reaction chamber (SCR) system (ultraWAVE™, Milestone, Sorisole, Italy) equipped with fifteen polytetrafluoroethylenes (PTFE) vessels (volume: 15 cm^3^ each).

#### 2.3.3. Sample Preparation

Samples were prepared by microwave acid digestion. The procedure allows the almost complete oxidation of the organic matrix and the extraction of metals and elements from the sample. Hence, optimal digestion was crucial for the ICP-MS analysis because it ensures recovery of all the analytes and bias-free preparation. The digestion procedure was performed using a mixture of 0.200 g of sample, directly weighed in the vessels, plus 1 mL HNO_3_ (67–69% (*w*/*v*)), 2 mL H_2_O_2_ (30% (*w*/*v*)), and 4 mL H_2_O. The digestion temperature of 240 °C was attained in 20 min and held for 10 min. After cooling at 40 °C, samples were collected, diluted to 15 mL, and filtered before analysis. Three blanks were prepared for each digestion batch, which underwent the same treatment as samples.

#### 2.3.4. Sample Analysis

The elemental analysis involved three different analytical techniques: ICP-MS, AAS, and UV–visible spectroscopy. ICP-MS is the ideal technique for trace and ultra-trace element analysis. On the other hand, AAS ensures excellent performances for the determination of elements in high concentrations, such as Na, Mg, K, and Ca. Finally, UV–visible spectroscopy is a valid alternative for phosphorus analysis, utilising the colorimetric method.

Concerning ICP-MS analysis, the elemental settings and parameters are reported in Table 1. This method was partially validated in terms of its limit of detection (LOD) and quantification (LOQ), repeatability (RDS%, analysis in triplicate), and linearity (calibration range). The validation parameters are reported in Table 2. Calibration standards were prepared from multi-elemental and single standard solutions in 2% HNO_3_ aqueous solution.

The elements found at very low or negligible concentrations in OP (e.g., rare earth elements) were not analysed in the fortified dough samples.

### 2.4. Dough Preparation

A type 00 commercial wheat flour (wheat variety: *Triticum aestivum*), provided by SIMEC Spa (Santa Giusta, Italy), was used for dough preparation. Its chemical composition was as follows: 14% moisture, 11.6% protein, 1.5% lipids, 3.8% fibre, and 0.1% ash. Dough samples were developed according to a basic bread formula: type 00 wheat flour, plus 1.8% salt, 2% yeast, and water. The amount of water to add was determined using a Farinograph-TS rheometer (model 827507, Brabender, Duisburg, Germany) equipped with a mixer S 300 N, and depended on the weight of wheat used and the substitution percentage of the ground FD-OP used (1%, 2%, 3%, and 5%), as shown in Table 3.

The dough for farinograph analysis was prepared using wheat flour, OP, and water only, to eliminate the effects that the salt or yeast may have on the results. Trials were repeated until an average consistency of 500 Brabender units was reached. A consistency between 480 and 520 was considered acceptable [44]. The pH of the dough was also measured before and after the first bulk leavening.

### 2.5. Dough Measurements

#### 2.5.1. Dough Extensibility and Stickiness

Uniaxial extensional characteristics of the experimental doughs were measured using a TA-XT plus texture analyser (Stable Micro Systems, Surrey, UK) with Texture Expert software 6.1.10.0 version for Windows with the Kieffer dough extensibility rig and a 30 kg load cell. A freshly prepared piece of dough weighing 30 g was placed in a Teflon mould with a grooved base that was lubricated with paraffin oil to prevent sample adhesion. The dough was prepared without the addition of yeast. The top plate of the mould was then attached to the dough to obtain uniform dough strips, which were then kept in a climate chamber set to 30 °C and 85% RH for 40 min before testing. The following settings were used to assess extensibility on at least ten dough strips: pre-test speed, 2.0 mm s^−1^; test speed, 3.3 mm s^−1^; post-test speed, 10.0 mm s^−1^; trigger force, 5 g; and an extensibility distance of 100 mm. Using the force–distance curves generated, we recorded the maximum peak force (in N), known as resistance to extension (REXT), and the distance (in mm) required to break the dough strips, noted as extensibility.

The stickiness of the dough was evaluated using a TA-XT plus texture analyser (Stable Micro Systems Ltd., Godalming, UK). To measure dough stickiness, a small amount of dough was placed in a chamber and closed firmly with a lid. The dough was then pushed out through small holes in the lid by turning an internal screw and left to rest for 30 s. To prevent loss of moisture, a perspex cap was placed over the dough surface. Afterwards, the cover was removed and a SMS/Chen-Hoseney dough stickiness rig (A/DSC) with a 25 mm cylindrical probe (probe SMS P/25) was connected to the load cell of the texture analyser. Measurements of dough viscosity (in N) were conducted ten times on each dough sample at room temperature, and each dough formulation was prepared and tested twice.

#### 2.5.2. Rheofermentometer Analysis

The fermentation characteristic of the doughs containing different percentages of FD-OP were assessed by measuring dough development and carbon dioxide production and retention using a Rheofermentometer F3 (Chopin, Villeneuve-La-Garenne, France). To perform the test, 315 g of dough was prepared in the same way as described above, then incubated for 180 min at a temperature of 28.5 °C in a fermentation chamber and covered with a stainless-steel cylinder without any additional weight. From the rheofermentometer curves, we obtained the following: the maximum dough development height (Hm, in mm); the time taken to achieve maximum dough rise (T1, in min); the final dough height at the end of the test (h, in mm); the percentage decrease in dough volume at the end of the test ((Hm-h)/Hm, in %); the maximum height of gaseous release (H’m, in mm); the total amount of gas produced (VTOT, in mL); the amount of gas retained (VRET, in mL); the amount of gas released (VREL, in mL); and the gas retention coefficient (RC), calculated as VRET/VTOT (%). The test was performed twice for each dough formulation.

#### 2.5.3. Small Deformation Characteristics

To determine the rheological properties of the dough, we performed an oscillation test using a dynamic shear rheometer (Anton Paar MCR 92, GmbH Inc., Graz, Austria) equipped with a 50 mm plate (P50/P2). The dough samples were prepared without yeast and left to rest at room temperature for 20 min before analysis. Around 2 g of dough core was taken using a spatula and placed between the plates for 2 min to relax, with the upper plate lowered against the sample to achieve a dough thickness of 2 mm. A thin layer of paraffin oil was applied to the sample edge to prevent moisture loss. To identify the region of linear viscoelasticity of the dough, we conducted a single amplitude oscillatory strain experiment (0.01–10%) on each sample applying an oscillation frequency of 1 Hz. The tests were performed at 20 °C. To measure storage modulus (G′) and loss modulus (G″), we made frequency sweep measurements (0.1–10 Hz) at a constant tension of γ 0.1% within the linear viscoelastic range. Each dough formulation was prepared twice, and each sample measured at least three time.

#### 2.5.4. Rapid Visco Analyser (RVA)

We used a rapid visco analyser (RVA-4, Newport Scientific, Warriewood, Australia) to analyse the pasting characteristics of the different dough samples (prepared using three different percentages of OP for each olive cultivar), and placed 3 g flour and 25 mL distilled water in the canister of the analyser to prevent the formation of lumps. The RVA test was conducted following the AACC method 76-21 [45]. The software program Thermocline 2.6 version for Windows was used to calculate the following parameters from the pasting curve: pasting temperature; peak time (when peak viscosity occurred); peak viscosity (maximum hot paste viscosity); holding viscosity (minimum hot paste viscosity); breakdown (peak viscosity minus holding viscosity); and total setback (final viscosity minus holding viscosity). Each dough formulation was prepared twice, and three measurements were recorded for each dough.

### 2.6. Statistical Analysis

Statistical analysis was performed using Minitab^®^ 17.1.0 software (Minitab, LLC, State College, PA, USA). All analyses were performed in at least triplicate. For OP analyses, *t*-tests were carried out to compare differences between cultivars. For dough analyses, means were compared by one-way analysis of variance (ANOVA), followed by Tukey HSD for the separation of significant means. Statistical significance was considered for a *p*-value < 0.05 (confidence level of 95%).

## 3. Results and Discussion

### 3.1. Characterization of Olive Pomace

The results of proximate analysis for raw and freeze-dried OP obtained from the two cultivars Bosana and Semidana are reported in Table 4 and Table 5, respectively; the colour analysis results are shown in Table 6, and the mineral compositions of FD-OP for the two cultivars are reported in Table 7.

OP is what remains after olive oil extraction. Its composition, which comprises water, oil, and an important quantity of bioactive compounds, depends on the olive variety, maturation stage, relative yield, climatic conditions, agricultural practices, the season, and storage conditions [46,47].

The macro composition of raw olive pomace (ROP) showed significant differences between the cultivars for all parameters analysed, except water activity. Bosana had a significantly higher dry matter (25.36%) content, as well as higher fat (17.57%), protein (6.14%), and ash (5.29%) content reported on dry matter. The OP from Bosana had a significantly greater protein content compared with that from Semidana, which also coincided with the nitrogen content that was used to calculate the protein content. As the moisture content of OP can vary greatly, reaching up to 70% [46], depending on the season, the variety, and the extraction system, it was necessary to obtain the dry matter content, which was below 30% for both varieties.

The results, in particular, the protein and nitrogen (N) content, agreed with those obtained in the study conducted by Nunes et al. [46]. Elemental analysis of FD-OP did not result in significant differences between cultivars in carbon (C) or hydrogen (H) content. Moreover, the results obtained for N, C, and H corroborated the findings from previous studies [48,49,50].

The freeze-drying process allowed us to obtain FD-OP for both cultivars with a moisture content lower than 5%. Caponio et al. [41] reported a similar dry matter and protein content for FD-OP, whereas significant differences were seen in the total dietary fibre (TDF)/DM, for which they reported 20.10% compared with the 61.1% (Bosana) to 66.7% (Semidana) observed in the present study. Other studies report TDF levels closer to those found here, such as 43.75% [51], 54.5% [52], and 57.96% [53].

The fat content corresponded to that reported by de Gennaro et al. [43], at around 22%.

Water activity (a_w_) did not exhibit any significant differences between cultivars and was considered relatively high, ranging from 0.97 to 0.98. These results were in accordance with those published by Alhamad et al. [54]. A high aw is attributed to the free movement of water molecules induced by the fat content [54]. Furthermore, such a high aw promotes bacterial growth in the matrix [54]; therefore, decreasing the water content by drying the pomace is recommended and promotes their antimicrobial activity related to the presence of antioxidants [55].

OP is known to be a good source of functional compounds, including polyphenols [41]. TPC significantly varied between the OP from Semidana, at 28.36 g GAE kg^−1^ FD-OP, and OP from Bosana, at 42.67 g GAE kg^−1^ FD-OP. Caponio et al. [41] reported a much higher TPC, higher with respect to our results at 75.76 g GAE kg^−1^ FD-OP. On the other hand, a lower TPC was reported by Simsek and Süfer [53], at 14.04 g GAE kg^−1^ FD-OP, and by Simonato et al. [39], at 25.23 g GAE kg^−1^ FD-OP. Simsek and Süfer [53] claimed that, during olive oil extraction, approximately 45% of the TPC is lost in the OP. The amount of polyphenols present in OP contributes to its distinguished antioxidant activity [43,56], but the DPPH assay in our study did not show any significant difference between the OPs obtained from the two cultivars studied. The literature reports similar results for the antioxidant activity of OP, as assessed by DPPH assay [41,43], but our results were distinctly higher than those reported by Simonato et al. [39] and Simsek and Süfer [53].

Regarding the colour attributes, the L* and a* parameters did not show any significant differences between the two cultivars, while the b* parameter was significantly different between the Bosana and Semidana OP, as shown in Table 6.

Figure 1 shows the photographs of freeze-dried OP from cultivars Bosana and Semidana. Cedola et al. [5] stated that the high content of polyphenols is responsible for the purplish colour of the OP, and this is explained by the negative b* value of the Bosana OP, which appears to be more purple. A lot of factors affect OP colour, but the proximate composition of the variety has the biggest influence [57].

To our knowledge, only a few studies have explored the mineral composition of OP from *Olea europaea* L., and none have conducted studies on Bosana or Semidana. OP is rich in minerals [58], and especially in macro elements important for the diet such as calcium, magnesium, sodium, potassium, and phosphorus (Table 7).

Only magnesium and potassium exhibited significant differences, and both were higher in FD-OP from the Bosana cultivar.

In a study conducted by Pošćić et al. [59], the elemental composition of OP was examined in order to improve the quality of EVOO, as residues of OP affect its overall composition and quality. The study examined OP from the cultivar Oblica, harvested along the Croatian coast, and all the detected elements fell within the same range as observed in the present study, with the exception of Na and P, which were higher in the OP of Bosana and Semidana.

The differences in the elemental composition of olives, their oils, and OP is greatly affected by the temperature, latitude, and distance from the sea of the provenance sites, as well as by the year of harvest, even at the same site [60].

### 3.2. Dough Preparation and Breadmaking %

#### The Pasting Profile of Dough

Table 8 reports the data on dough development time (DDT), consistency, water added (WA), and stability (S) for the control dough and the doughs obtained through the partial substitution of flour with different percentages of Bosana and Semidana OP. 

Water absorption is affected by the dough’s starch, damaged starch, and protein and pentosan content, and by its gluten network. Freeze-dried ground OP has a softer texture than wheat flour, and this allows the OP particles to hydrate and swell faster than those of the wheat flour, necessitating a smaller percentage of added water in the fortified doughs [44,61]. As reported in Table 3, the added water percentage became lower as the OP inclusion rates rose, starting at 54% for 1% Bosana, and 54.5% for 1% Semidana. This may be due to the fact that, although the substituting pomace percentage was low, its presence increased the protein content in the dough, but it was not high enough to decrease the starch content to a point that the presence of both protein and starch required these water absorption percentages [61].

The time required to form the control dough was around 17 min. However, the addition of OP significantly quickened dough formation, which took a maximum of 1 min 30 sec to make (Table 8). The DDT is the length of time between the addition of water and the beginning of the formation of a homogenous and smooth dough. The doughs fortified with FD-OP had a better water absorption capacity, associated with the prominent fibre content, which led to better consistency and faster dough formation, thus decreasing the DDT [44,62].

The variable ‘stability’ showed the same trend as the added water percentage, and it was significantly ameliorated with the addition of Semidana FD-OP at 1%, while it remained the same as for the control dough upon the incorporation of Bosana FD-OP at 1% and Semidana FD-OP at 2% and 3%, which showed non-significant differences between them. As reported in Table 8, the stability time of dough at these FD-OP incorporation levels varied between 2 min 1 s and 2 min 39 s; the stability time of the control dough was 2 min. The stability time is the duration of mixing at which the dough has an ideal consistency. The higher stability of the dough with lower pomace substitution percentages is associated with the interactions between the dietary fibre, water, and flour protein, which disrupt the gluten–starch interface, leading to better stability [44,62]. By contrast, the lower stability of the dough with higher OP percentages was related to their lower gluten content and, thus, weaker gluten network, as a consequence of the lower starch percentage; the weaker gluten network was also favoured by the ability of OP to soften the dough [44,62].

### 3.3. Dough Measurements

#### 3.3.1. Extensibility and Stickiness of the Dough

Dough stickiness, also known as dough adhesiveness, refers to the dough’s ability to adhere to surfaces. Higher stickiness values are not favourable because increased adhesiveness may result in production interruption, waste, and contamination. Likewise, low stickiness is also not desirable as it would produce a dough that is unable to maintain its shape [63,64]. The stickiness of the control dough versus the doughs with different OP substitution levels are summarised in Table 9.

The incorporation of FD-OP at 1% for both *cultivars* and at 2% for the Semidana cultivar had no significant effect on dough stickiness. However, all the other percentages significantly decreased the stickiness of their respective doughs. Higher stickiness levels are attributed to higher amounts of water which are distributed in the dough either in bulk or within the starch–gluten matrix. Free water molecules are responsible for the stickiness. Therefore, increasing fibre content in the dough, by increasing the OP substitution level, traps water molecules within the formed fibre connections, disrupting the gluten network. As a result, water molecule availability decreases and the dough’s stickiness also decreases [63,64]. Similar results were obtained by Struck et al. after fortifying bread dough with blackcurrant pomace, which is also rich in fibre. [64].

Measurements of the dough’s resistance to extension are reported in Table 9. Increasing the FD-OP substitution level led to a significant increase in the dough’s resistance to extension at all the inclusion percentages compared with the control, as well as between inclusion levels. The effects of FD-OP on the dough’s extensibility properties, which became less flexible and more rigid with respect to the control dough, were consistent with the literature and other research using different sources of dietary fibre such as rosehip seed flour [8], *Malva aegyptiaca* L. leaf powder [65], brewer’s spent grain, and apple pomace [66]. In accordance with Gül and Şen [8] and Struck et al. [64], these results can be attributed to poorer gluten networks due to fibre incorporation, resulting in stiffer and firmer dough. Glutenin is the building block of the gluten network and it is responsible for the dough’s elasticity. It achieves this through creating intermolecular disulphide bonds with other glutenin molecules [67,68]. These cross-links create a stable and elastic network, which gives the dough its resistance [67,68]. This explains the decrease in elasticity upon dilution of the gluten content with fibre-rich FD-OP.

#### 3.3.2. Rheofermentometer Analysis

The dough fermentation properties, as measured by rheofermentometry, are shown in Table 10.

Upon fermentation, yeast produces gas that spreads into the dough and increases the number of air bubbles. The withheld gas in the dough is the ratio between gas production and gas retention. On the other hand, a lower (Hm-h)/Hm indicates greater dough stability and, therefore, a greater gas retention ability [63].

Increasing the OP percentages quickened dough formation. The sample height (Hm) of all samples fortified with OP, from both olive cultivars and at all percentages (1%, 2%, 3%, and 5%), were significant lower compared with the control. The Hm decreased as the percentage of OP substitution increased. This is because ground FD-OP is both gluten-free and fibre-rich, which, subsequently, dilutes and, thus, disrupts the dough’s gluten network, leading to a less-risen dough. A lower Hm denotes that the gas produced during fermentation was not sufficient or properly distributed within the air bubbles to support the overall structure of the dough as it should have during proofing. This can result in a dough that does not rise properly, leading to a denser bread that is not as light and airy as desired [4]. Consequently, increasing the FD-OP levels decreased gas production (Vt) due to the reduced particle size of the FD-OP and the reduced interactions of bioactive compounds found in the FD-OP with the activity of the yeast responsible for gas production. This decrease in gas production is inversely proportionate to the gas retention capacity expressed by Vr/Vt [4]. This explains the significant increase in gas retention with the addition of the OP, regardless of the cultivar and the percentage of use, and it highlights the greater ability of the OP-fortified dough samples to retain gas during the leavening process, as reflected in the significantly higher gas retention capacities for both cultivars compared with the control (*p*-value < 0.05).

The dough’s stretching capacity is also associated with the retention coefficient, as well as with the quality of the gluten protein network and particle size [4,63]. This means that increasing the FD-OP percentage progressively weakens the gluten network, and produces a dough with a poorer capacity to increase in volume, but FD-OP dough also has less gas production and a smaller particle size, equating to better gas retention. However, apart from the control, the best dough consistency, viscosity, and stability values were found in the dough with 1% Semidana FD-OP. This dough showed a greater water absorption capacity. Higher percentages of FD-OP resulted in reduced dough consistency.

#### 3.3.3. Small Deformation Characteristics (Rheometer Analysis)

The storage modulus (G’) and loss modulus (G”) of the dough samples are reported in Figure 2.

G′ represents the “solid-like” behaviour of the matrix. It measures the stored elastic energy following the application of a shear stress and is calculated as the ratio of stress to in-phase strain [69,70].

G″ represents the “liquid-like” behaviour of the matrix. It measures the dissipation of energy as heat viscously and is calculated as the ratio of the viscous out-of-phase component to shear stress [69,70].

In all samples, G′ was greater than G″, as shown in all parts of Figure 2, and both moduli increased as the frequency increased. Having a G′ that is greater than G″ means that the dough’s consistency is semi-solid and elastic—the necessary consistency for bread dough [2,63,71].

Moreover, at all frequencies, both the storage modulus and the loss modulus increased with increasing levels of OP inclusion, meaning that the dough viscoelasticity also increased. This occurred due to the presence of fibre and polyphenols in the OP. In brief, proteins in the dough form complexes through hydrogen bonds between their carbonyl group and a hydroxyl group on the polyphenols, and this enhances the gluten network, increasing their G’ and, thus, their elasticity, and hence explaining the increase in G’ with increasing levels of OP inclusion. Additionally, fibre and fibre-like structures can make temporary connections between each other, thus disrupting the gluten network due to the competition for water absorption as gluten makes water less available. Fibre connections serve as ‘fillers’ in the dough, rendering it thicker, and, therefore, augmenting its viscous properties and explaining the increase in G″ with increasing levels of OP inclusion [2,4]. These results are in accordance with others reported in literature [2,4,63].

#### 3.3.4. Rapid Visco Analyser (RVA)

Figure 3 demonstrates the effect of different FD-OP inclusion percentages on the pasting properties of dough.

Peak viscosity is the maximum viscosity achieved during the heating phase. It indicates the ability of starch to absorb and hold water, reducing the presence of free water molecules. It reflects the swelling and gelatinization of starch granules—known as pasting. The ‘holding strength’, also known as the ‘trough’, is the ability of the dough to hold its viscosity after reaching the peak and before it cools down. The ‘breakdown’ is the difference in viscosity between the peak and trough. The ‘final viscosity’ is that achieved at the end of the test, and the ‘setback’, the increase of viscosity during the cooling phase, is calculated by subtracting the trough viscosity from the final viscosity [63].

The values of peak 1, final viscosity, trough 1 (holding strength), and final viscosity were all higher in the control, in agreement with the findings reported by Farbo et al. [63]. The addition of OP of both cultivars reduced these parameters at all inclusion rates with respect to the control dough, although not to a significant degree at the 1% inclusion rate. Fortified doughs had significantly higher values of breakdown, that is, the reduction in viscosity during the holding phase, compared with the control dough. The higher the OP percentage, the higher the breakdown. The control dough, which does not contain any OP, has a greater starch content than the fortified doughs, enabling it to exhibit the best viscosity. Specifically, the incorporation of OP decreases the gluten content of the dough and increases its water absorption capacity, as mentioned previously, which leads to a decrease in the amount of the swollen starch granules contributing to the viscosity, thus decreasing the viscosity value [62].

Another factor which may contribute to these results is the increase in the activity of the alpha-amylase naturally present in wheat flour due to the increase in calcium content with increasing OP substitution levels. Alpha-amylase depends on calcium ions (Ca^2+^) for enhanced activity [4]. Thus, increasing levels of starch hydrolysis with increasing OP inclusion would also have contributed to decreased viscosity with respect to the control dough.

## 4. Conclusions

The present work demonstrates that incorporating FD-OP into wheat dough at increasing flour substitution percentages (1, 2, 3, and 5%) significantly modifies the properties of dough. This primarily occurs due to the high fibre content of OP compared with type 00 wheat flour (which contains no fibre) and the high abundance of polyphenols in OP. Low substitution levels of OP improved water absorption and decreased dough development time, resulting in faster dough formation (17:40 min in controls vs. values < 1:30 min in FD-OP fortified dough). In addition, compared with the control dough, the stability was improved in dough fortified with 1% Bosana, 1% Semidana, and 2% Semidana FD-OP, whereas greater levels of substitution retrograded the stability of the dough. Moreover, the storage (G′) and loss (G″) moduli showed increasing trends as FD-OP levels increased, which strengthened the viscoelastic properties of the dough. FD-OP inclusion decreased dough stickiness and elasticity, except at the 1 and 2% substitution levels. During fermentation, gas retention was significantly higher in doughs fortified with FD-OP at all levels: the control dough retained 90.90% of the gas, whereas the lowest level of gas retention in the fortified doughs was 95.90%. Thus, we can safely conclude that the incorporation of FD-OP at 1% Bosana and 1 and 2% Semidana significantly improved all rheological properties of the wheat dough due to the presence of fibre and polyphenols, while higher substitution percentages had an adverse effect because of the significant disruption to the gluten network, resulting in a gluten-free-like dough.

Future work is required to study the effects of OP inclusion on breadmaking, including the resulting nutritional, physicochemical, and rheological profiles of bread. Considering the results of the present study, a substitution level of 5% FD-OP can be discarded from further analyses.

## Figures and Tables

**Figure 1 foods-13-00478-f001:**
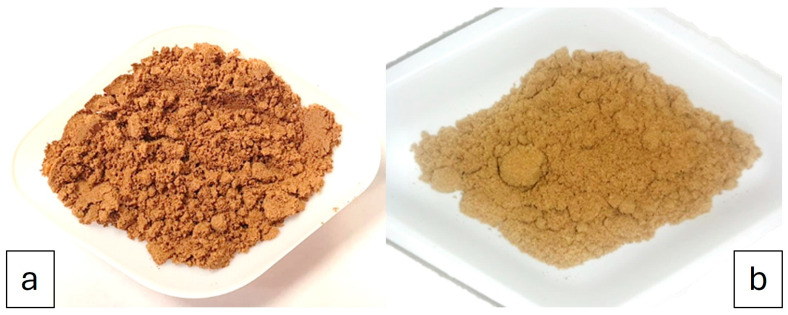
Digital photograph of FD-OP from Bosana (**a**) and Semidana (**b**) cultivars.

**Figure 2 foods-13-00478-f002:**
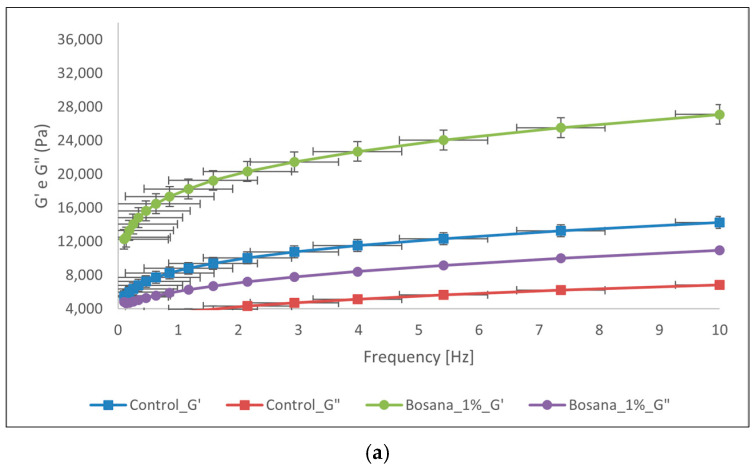
Dynamic rheological parameters: storage modulus (G′) and loss modulus (G″) of the control dough compared with the dough fortified with different percentages of OP as a function of frequency. (**a**) Bosana 1% dough vs. control; (**b**) Bosana 2% dough vs. control; (**c**) Bosana 3% dough vs. control; (**d**) Bosana 5% dough vs. control; (**e**) Semidana 1% dough vs. control; (**f**) Semidana 2% dough vs. control; (**g**) Semidana 3% dough vs. control; and (**h**) Semidana 5% dough vs. control.

**Figure 3 foods-13-00478-f003:**
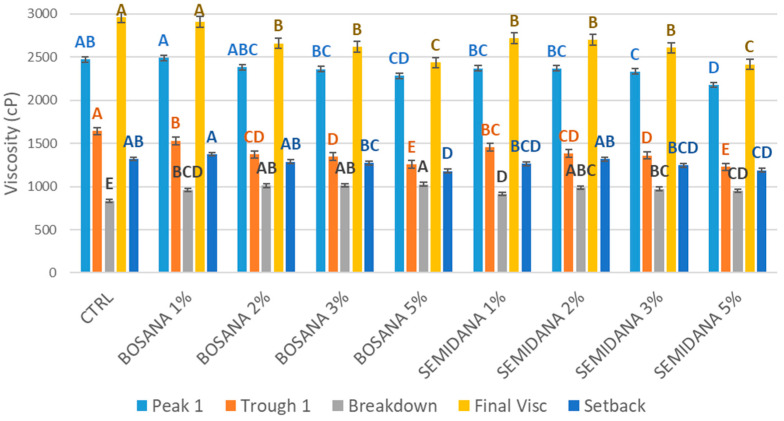
Effect of different OP inclusion percentages on the pasting properties of dough, peak viscosity (peak), holding strength (trough), the difference between the peak and trough (breakdown), final viscosity, and the difference between final viscosity and trough (setback). Error bars indicate the standard error. Different letters within the same group of graph bars indicate significantly different means as assessed by one-way ANOVA, followed by Tukey HSD (*p*-value < 0.05).

**Table 1 foods-13-00478-t001:** Instrumental parameters and elemental settings of NexION 300X ICP-MS.

ICP-MS NexION 300X Perkin Elmer Settings			
RF power generator (W)	1300	KED mode cell entrance voltage (V)	−8.0
Ar plasma flow (dm^3^ min^−1^)	18.0	KED mode cell exit voltage (V)	−25.0
Ar auxiliary flow (dm^3^ min^−1^)	1.20	Resolution (Da)	0.7
Ar nebuliser flow (dm^3^ min^−1^)	0.91	Scan mode	Peak hopping
Nebuliser	Meinhardt^®^, glass	Detector mode	Dual
Spray chamber	Cyclonic, glass	Dwell time (ms)	50
Skimmer and sampling cones	Nickel	Number of points per peak	3
Sampling depth (mm)	0	Acquisition time (s)	6
Deflector voltage (V)	−8.00	Acquisition dead time (ns)	35
Analog stage voltage (V)	−1750	KED gas	Helium 99.999%, flow 4.2 cm^3^ min^−1^
Pulse stage voltage	+1350	Masses of optimization	^7^Li, ^115^In, and ^205^Tl

**Table 2 foods-13-00478-t002:** Validation parameters of the ICP-MS method.

Element	Isotope	Abundance (%)	Analysis Mode	Linearity (R^2^)	Calibration Range	LOD (µg kg^−1^)	LOQ (µg kg^−1^)	Repeatability (RSD%)
Li	7.016	92.50	STD	0.99976	0.1–100	0.844	2.785	5%
Rb	84.912	72.17	STD	0.99996	0.1–200	0.429	1.415	5%
Sr	87.906	82.58	STD	0.99994	0.1–200	0.146	0.483	5%
Cs	132.905	100.00	STD	0.99985	0.1–100	0.108	0.355	4%
Ba	137.905	71.70	KED	0.99994	0.1–200	0.194	0.642	4%
Pb	207.977	52.40	STD	0.99985	0.1–100	0.009	0.031	4%
Bi	208.980	100.00	STD	0.99986	0.01–10	0.019	0.061	6%
Sc	44.956	100.00	KED	0.99998	0.1–100	1.302	4.297	21%
V	50.944	100.00	KED	0.99998	0.1–100	0.764	2.522	3%
Cr	51.941	83.79	KED	0.99999	0.1–100	1.935	6.386	1%
Mn	54.938	100.00	KED	0.99997	0.1–200	6.252	20.632	1%
Fe	56.935	2.20	KED	0.99984	0.1–200	29.264	96.570	1%
Co	58.933	100.00	KED	0.99998	0.1–200	1.870	6.171	2%
Ni	59.933	26.23	KED	0.99997	0.1–200	1.370	4.522	1%
Cu	62.930	69.17	KED	0.99998	0.1–200	1.342	4.429	1%
Zn	65.926	27.90	KED	0.99996	0.1–200	7.605	25.095	1%
As	74.922	100.00	KED	0.99999	0.1–100	3.081	10.166	11%
Se	81.917	8.73	KED	0.99961	0.1–200	2.587	8.536	17%
Y	88.905	100.00	STD	0.99995	0.1–100	2.074	6.845	5%
Mo	97.906	24.13	STD	0.99996	0.01–10	0.528	1.742	2%
Cd	110.904	12.80	STD	0.99995	0.01–10	2.245	7.408	7%
Sn	117.902	24.22	STD	0.99996	0.01–10	0.711	2.347	3%
Sb	120.904	57.36	STD	0.99995	0.01–10	1.627	5.368	6%
La	138.906	99.91	KED	1.00000	0.01–10	0.715	2.358	1%
Ce	139.905	88.45	KED	1.00000	0.01–10	3.291	10.860	1%
Pr	140.907	100.00	KED	1.00000	0.01–10	0.473	1.560	2%
Nd	141.908	27.20	KED	1.00000	0.01–10	3.550	11.714	7%
Sm	151.920	26.75	KED	1.00000	0.01–10	0.515	1.698	6%
Eu	152.929	52.19	KED	1.00000	0.01–10	0.117	0.386	14%
Gd	157.924	24.84	KED	1.00000	0.01–10	0.271	0.896	4%
Tb	158.925	100.00	KED	1.00000	0.01–10	0.072	0.238	14%
Dy	163.929	28.18	KED	1.00000	0.01–10	0.243	0.802	10%
Ho	164.930	100.00	KED	1.00000	0.01–10	0.084	0.279	4%
Er	165.930	33.61	KED	1.00000	0.01–10	0.114	0.378	6%
Tm	168.934	100.00	KED	1.00000	0.01–10	0.082	0.270	5%
Yb	173.939	31.83	KED	1.00000	0.01–10	0.133	0.439	5%
Lu	174.941	97.41	KED	1.00000	0.01–10	0.108	0.356	20%
Tl	204.975	70.48	STD	1.00000	0.01–10	0.211	0.697	2%
Hg	201.971	29.86	STD	0.99752	0.1–100	0.345	1.139	5%
U	238.050	99.28	STD	0.99981	0.01–10	0.011	0.036	3%

**Table 3 foods-13-00478-t003:** Bread dough formulations: percentage of FD-OP, yeast, and salt for different water percentages as determined by farinography.

Sample	Water (%)	Yeast (%)	Salt (%)
Control (CTRL)	54.0	2	1.8
Bosana FD-OP (1%)	54.0	2	1.8
Bosana FD-OP (2%)	53.5	2	1.8
Bosana FD-OP (3%)	53.0	2	1.8
Bosana FD-OP (5%)	51.8	2	1.8
Semidana FD-OP (1%)	54.5	2	1.8
Semidana FD-OP (2%)	54.2	2	1.8
Semidana FD-OP (3%)	53.5	2	1.8
Semidana FD-OP (5%)	53.0	2	1.8

**Table 4 foods-13-00478-t004:** Macro composition of raw olive pomace (ROP) from olive cultivars Bosana and Semidana.

Samples	Dry Matter % (DM) ROP	Fat/DM % ROP	Protein/DM % ROP	Ash/DM % ROP	Water Activity aw (ROP)
Bosana	25.36 ^a^ ± 0.22	17.57 ^a^ ± 0.16	6.14 ^a^ ± 0.11	5.29 ^a^ ± 1.05	0.98 ^a^ ± 0.01
Semidana	19.67 ^b^ ± 1.30	12.5 ^b^ ± 1.22	5.29 ^b^ ± 0.09	4.29 ^b^ ± 0.33	0.97 ^a^ ± 0.01

Each value represents the mean ± SD. Different superscripts within the same column indicate significant differences as assessed by *t*-tests (*p*-value < 0.05).

**Table 5 foods-13-00478-t005:** Macro composition of freeze-dried olive pomace (FD-OP) from olive cultivars Bosana and Semidana.

Samples	DM% FD-OP	TDF/DM % FD-OP	C/DM% FD-OP	H/DM% FD-OP	N/DM% FD-OP	TPC (g GAE kg^−1^ DM FD-OP)	DPPH (µmol TE g^−1^ DM FD-OP)
Bosana	98.49 ^a^ ± 0.07	61.1 ^b^ ± 0.99	56.23 ^a^ ± 1.60	7.68 ^a^ ± 0.21	0.98 ^a^ ± 0.02	42.67 ^a^ ± 1.14	227.07 ^a^ ± 90.86
Semidana	98.31 ^a^ ± 0.46	66.7 ^a^ ± 0.52	54.51 ^a^ ± 0.35	7.19 ^a^ ± 0.07	0.85 ^b^ ± 0.01	28.36 ^b^ ± 1.73	195.36 ^a^ ± 70.93

Each value represents the mean ± SD. Different superscripts within the same column indicate significant differences as assessed by *t*-tests (*p*-value < 0.05).

**Table 6 foods-13-00478-t006:** Colour analysis of FD-OP from cultivars Bosana and Semidana.

Sample	L*	a*	b*
Bosana	98.17 ^a^ ± 0.92	0.33 ^a^ ± 0.19	−2.45 ^a^ ± 1.46
Semidana	98.68 ^a^ ± 5.46	−0.11 ^a^ ± 1.19	1.88 ^b^ ± 2.38

Each value represents the mean ± SD. Different superscripts within the same column indicate significant differences as assessed by *t*-tests (*p*-value < 0.05).

**Table 7 foods-13-00478-t007:** Mineral composition of freeze-dried olive pomace (mg kg^−1^ FD-OP) from cultivars Bosana and Semidana.

FD-OP Cultivar	Calcium (Ca)	Magnesium (Mg)	Sodium (Na)	Potassium (K)	Phosphorus (P)
Bosana	830.8 ^a^ ± 34.3	601.5 ^a^ ± 2.1	458.6 ^a^ ± 9.0	13,269.9 ^a^ ± 431.5	1353.3 ^a^ ± 48.4
Semidana	759.4 ^a^ ± 8.1	526.3 ^b^ ± 5.6	432.4 ^a^ ± 20.5	11,577.9 ^b^± 195.9	1466.0 ^a^ ± 37.6

Each value represents the mean ± SD. Different superscripts within the same column indicate significant differences as assessed by *t*-tests (*p*-value < 0.05).

**Table 8 foods-13-00478-t008:** Dough mixing properties according to the farinograph.

Sample	Dough Development Time (min:s)	Consistency (FE)	Added Water % (*w*/*w*)	Stability (min:s)
CTRL	17:40 ^a^ ± 00:59.4	502.0 ^a^ ± 2.8	54.0	2:00 ^b^ ± 00:02.83
1% Bosana	01:25 ^b^ ± 00:01.41	500.0 ^a^ ± 4.2	54.0	2:01 ^b^ ± 00:08.49
2% Bosana	01:28 ^b^ ± 00:02.83	494.5 ^a^ ± 6.4	53.5	1:32 ^c,d^ ± 00:02.83
3% Bosana	01:12 ^b^ ± 00:04.24	497.5 ^a^ ± 4.9	53.0	1:17 ^d^ ± 00:07.07
5% Bosana	01:14 ^b^ ± 00:07.07	493.0 ^a^ ± 2.8	51.8	1:10 ^d^ ± 00:01.41
1% Semidana	01:30 ^b^ ± 00:04.24	498.0 ^a^ ± 0.0	54.5	2:39 ^a^ ± 00:09.90
2% Semidana	01:19 ^b^ ± 00:05.66	497.5 ^a^ ± 7.8	54.2	2:02 ^b^ ± 00:11.31
3% Semidana	01:26 ^b^ ± 00:01.41	494.5 ^a^ ± 4.9	53.5	1:54 ^b,c^ ± 00:02.83
5% Semidana	01:19 ^b^ ± 00:04.24	487.5 ^a^ ± 3.5	53.0	1:27 ^d^ ± 00:04.24

Each value represents the mean ± SD. Different superscripts within the same column indicate significant differences as assessed by one-way ANOVA, followed by Tukey HSD (*p*-value < 0.05).

**Table 9 foods-13-00478-t009:** Effect of OP on dough stickiness and resistance to extension.

Sample	Stickiness (N)	Resistance to Extension (N)
CTRL	0.34 ^ab^ ± 0.02	0.11 ^f^ ± 0.00
1% Bosana	0.36 ^a^ ± 0.03	0.34 ^bc^ ± 0.03
2% Bosana	0.24 ^d^ ± 0.01	0.27 ^b^ ± 0.02
3% Bosana	0.19 ^e^ ± 0.01	0.30 ^cd^ ± 0.04
5% Bosana	0.15 ^f^ ± 0.02	0.44 ^a^ ± 0.03
1% Semidana	0.34 ^ab^ ± 0.02	0.24 ^e^ ± 0.04
2% Semidana	0.35 ^ab^ ± 0.02	0.27 ^de^ ± 0.03
3% Semidana	0.32 ^b^ ± 0.03	0.32 ^c^ ± 0.03
5% Semidana	0.28 ^c^ ± 0.03	0.33 ^bc^ ± 0.04

Each value represents the mean ± SD. Different superscripts within the same column indicate significant differences as assessed by one-way ANOVA, followed by Tukey HSD (*p*-value < 0.05).

**Table 10 foods-13-00478-t010:** Effect of Bosana and Semidana FD-OP on the leavening properties of dough as assessed through rheofermentometry.

Sample	Dough Development			Gas Behaviour	
	Hm (mm)	(Hm-h)/Hm (%)	T1(h)	T’1 (h)	(CR) Vr/Vt: (%)
CTRL	51.10 ^a^ ± 0.85	5.50 ^b,c^ ± 2.69	01:47:15	01:54:45	90.90 ^c^ ± 0.71
1% Bosana	31.95 ^c^ ± 0.07	1.00 ^d^ ± 0.14	02:35:15	02:16:30	95.90 ^b^ ± 3.25
2% Bosana	23.75 ^d^ ± 0.35	3.65 ^c^ ± 0.07	02:39:00	02:27:45	98.80 ^a^ ± 0.00
3% Bosana	13.00 ^e^ ± 3.96	8.40 ^a^ ± 0.42	02:24:00	02:26:15	99.10 ^a^ ± 0.14
5% Bosana	12.55 ^e^ ± 0.07	6.05 ^b^ ± 1.34	01:46:30	02:30:45	99.20 ^a^ ± 0.14
1% Semidana	41.40 ^b^ ± 0.57	0.65 ^d^ ± 0.92	02:48:00	02:23:00	98.85 ^a^ ± 0.07
2% Semidana	32.10 ^c^ ± 1.13	0.30 ^d^ ± 0.28	02:57:45	02:30:45	98.95 ^a^ ± 0.21
3% Semidana	25.10 ^d^ ± 6.90	0.50 ^d^ ± 0.25	02:53:00	02:35:30	99.16 ^a^ ± 0.21
5% Semidana	19.50 ^de^ ± 0.28	0.80 ^d^ ± 0.85	02:51:45	02:38:15	99.20 ^a^ ± 0.14

Hm is the maximum dough development height; (Hm-h)/Hm represents the decrease in dough height at the end of the 3 h analysis time; and T1 is the time at which the dough reaches its maximum height. In gas behaviour, T’1 is the time for maximum gas production; and (CR) Vr/Vt is the percentage of the gas withheld at the end of the 3 h analysis period [63]. Different letters within the same column indicate significant differences as assessed by one-way ANOVA, followed by Tukey HSD (*p*-value < 0.05).

## Data Availability

Data is contained within the article.

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
