# Peer review of "Improving the Rheological Properties of Dough Obtained by Partial Substitution of Wheat Flour with Freeze-Dried Olive Pomace"

_foods, 2024, doi:10.3390/foods13030478_

Round 1

Reviewer 1 Report

Comments and Suggestions for Authors

Comments for Authors

The authors conducted a study entitled "Improving the rheological properties of dough obtained by partial substitution of wheat flour with freeze-dried olive pomace", which aimed to extract olive pomace from extra virgin olive oil production and incorporate in wheat flour. Furthermore, the effect of incorporating olive pomace into wheat dough was investigated. It is interesting and useful to use agro-food wastes as a by-product in food to produce sustainable and nutritious products.

The authors have put an effort into providing explanations and have commented on previous findings of the international literature.Top of Form The results of this study is meaningful and interesting. However, there still many issues should be resolved in the present version.

Abstract

The results presented in the abstract do not correspond to the results indicated in the tables and the discussion. Please add findings in the abstract section.

Line 30-31, Why aren't the keywords in alphabetical order? Organizing keywords alphabetically enhances clarity and facilitates easier navigation for readers and researchers, ensuring a more user-friendly and efficient search experience.

Introduction

Line 49-50, Include a recent study of https://onlinelibrary.wiley.com/doi/full/10.1002/fsn3.2323;  which incorporates corn fiber (bran) into cereal-based food products.

The introduction lacks a clearly stated reason for writing this scientific study, a really clearly stated main goal and secondary goals.

Describe more about functional and rheological properties of fiber addition if the functional products are so challenging.

Line 85-89, Authors have not mentioned a clear rationale for this study design, why they use olive pomace.

Line 88,89, “cereal-based foods, namely bread” change with “cereal-based food products.”

Materials and Methods

Line 110-111, Add method number of fat/lipid content determination from Association of Official Agricultural Chemists (AOAC).

In line 136, which reagent was used to acidify the sample?

Line 299-300, The RVA test was conducted following AACC 76-21, please correct it as The RVA test was conducted following the AACC method 76-21 [43].

Results

Line 338-341, rephrase the sentence for better understanding.

Line 497-499, The height (Hm) of all samples fortified…., rewrite the statement for clarity.

The paragraph should be rewritten to remove grammatical errors and syntax mistakes between lines 516-523.

Conclusion

It is important that conclusions focus on the overall outcome of the study and are more targeted. The present section is a summary of the results. The conclusion should include numerical values along with mean findings.

Overall, the current research paper should be proofread to correct grammatical errors, sentence structure errors, and incomplete thoughts.

Comments on the Quality of English Language

Extensive editing of English language required

The current research paper should be proofread to correct grammatical errors, sentence structure errors, and incomplete thoughts.

Author Response

Dear Reviewer,

thank you very much for your review. In the attached file you can find all our replies.

Best regards

Antonio Piga

Reviewer 2 Report

Comments and Suggestions for Authors

In the study, the authors analyzed the impact of olive pomace addition on the rheological properties of wheat dough. In the introduction, they correctly addressed issues related to the potential use of by-products for food enrichment and characterized the properties of olive pomace. The research methods are appropriate, and the results were adequately analyzed, compiled, and discussed. However, I have a few comments on the paper:

  1. What were the conditions of olive pomace lyophilization (temperature and pressure in the lyophilizer chamber)?

  2. What was the particle size of the ground pomace used in the study?

  3. While the authors provided a detailed characterization of the olive pomace used in the research, they did not present the characteristics of the wheat flour. At least basic features such as chemical composition and enzymatic activity (falling number) should be included in the paper.

  4. Regarding Figure 2, please provide a more precise explanation of the legend under the figure (2a, 2b... 2h).

  5. The authors are encouraged to calculate the coefficient of correlation between the additive level and the determined parameter(s).

Author Response

Dear Reviewer,

thank you very much for your revision. In the attached file you can find all our replies.

Best regards

Antonio Piga

Round 2

Reviewer 2 Report

Comments and Suggestions for Authors

The authors corrected the manuscript accordingly.